# OpenReview forum: "Scaling Laws For Scalable Oversight"
_NeurIPS.cc/2025/Conference — NeurIPS 2025 spotlight_

### Official Review · Reviewer_WwRt · 2025-06-28

**Clarity:** 4
**Significance:** 3
**Originality:** 4
**Rating:** 5
**Confidence:** 4

**Summary:**

This paper tests scaling laws for scalable oversight: how effective are weak supervisors at supervising strong models on hard tasks, and how does this effectiveness scale with the overall strength of the models (both supervisor and policy model)? This is evaluated on several game domains using general Elo scores to represent overall model strength.

**Questions:**

- Can you say something about how these results might vary when you have models that have stronger e.g. generative/discriminative gaps, and how we might aim to bring that to be the case?
- You have some recommendations in the paper, e.g., about intercept and slope of the scaling trends that should be ideal for a scalable oversight method. Can you say more in the paper about how that might be done, and perhaps provide a consolidated set of recommendations for future work at the end of the paper?

**Ethical Concerns:**

["NO or VERY MINOR ethics concerns only"]

**Limitations:**

yes (besides anything I'm bringing up in weaknesses)

**Quality:**

3

**Strengths And Weaknesses:**

Strengths:
- Applying the scaling laws lens to scalable oversight is a great idea and I think this is a fruitful direction to explore
- The Elo modeling approach as a general model for capabilities and deriving scaling laws is simple and sensible, if somewhat reductive
- The paper does come out with interesting insights and raises interesting concerns about how scalable oversight will scale

Weaknesses:
- The data here is mostly toy/synthetic and I think that makes it hard to extrapolate to real use cases for scalable oversight. The challenge with scalable oversight right now is really on the data side—how do we test it, and how do we implement it in practice with AI assisting humans, etc.? Using synthetic data avoids these concerns which can be good for getting simple/clear results but risks losing external validity. TBF, the debate experiments use more realistic data, but unless I'm mistaken, these are mostly redundant with the Khan et al and Kenton et al debate results, which also demonstrated scaling trends. The backdoor experiments also are a tad more realistic, if not entirely.
- More concretely, I think where the rubber meets the road in scalable oversight is around models' actual skill profile in validation and reward modeling. Scalable oversight can work if we can come up with a way to decompose and reframe problems so that a reward model that is good enough at specific skills related to verification can very robustly evaluate a policy for performing a task. That's what will determine scaling laws in practice and it's hard to get at that in toy settings.
- Minor but "scalable oversight" is usually more generally defined as involving methods for weak or resource-scarce supervisors to provide good feedback signal to models performing hard tasks. This includes eg developing methods for humans to supervise powerful AI. "weaker AI systems supervising stronger ones" is a sub-case of scalable oversight though.
- I think the NSO theory is great but more needs to be said about how 'weak models supervising strong models' is reductive in collapsing a complex skill profile into a single measure of intelligence. It's reasonable when applied to a single task though.

All that said, I think applying the scaling laws lens here is fruitful and these experiments are a nice step.

---

> ### Author Rebuttal · Authors · 2025-07-28
>
> Thank you for your careful and excellent comments! We are especially grateful for your thoughts on including recommendations for future work, and we are excited that you found our insights interesting and believe this is a fruitful direction. Please let us know if you have any remaining questions.
>
> > More concretely, I think where the rubber meets the road in scalable oversight is around models' actual skill profile in validation and reward modeling. Scalable oversight can work if we can come up with a way to decompose and reframe problems so that a reward model that is good enough at specific skills related to verification can very robustly evaluate a policy for performing a task. That's what will determine scaling laws in practice and it's hard to get at that in toy settings.
>
> Thank you for your comments about realism, these are good points! We agree that there is definitely still a non-trivial distance between our results and the real world. However, we believe our work is still useful in so far that it shows that many tasks do follow our framework. We thus establish that it is plausible that many real world tasks do indeed follow scaling laws like our framework, and the analysis here makes it easier to determine best oversight practices once we do fit tasks to a scaling law.
>
> > Minor but "scalable oversight" is usually more generally defined as involving methods for weak or resource-scarce supervisors to provide good feedback signal to models performing hard tasks. This includes eg developing methods for humans to supervise powerful AI. "weaker AI systems supervising stronger ones" is a sub-case of scalable oversight though.
>
> Thank you for this comment! This is correct, our work is a subset focused on the control problem, where we test a weaker agent’s ability to detect deceptive ability in a stronger model. In future work, we would like to adopt a setting similar to more typical weak to strong generalization [1], where we finetune a stronger model on “values” RLHF’d into a weaker model, and continue the hierarchical finetune with stronger and stronger models. Additionally, we think a relevant setting would be using weaker models in alignment audits [2] to detect misaligned behaviors in stronger models.
> [1] Weak-to-Strong Generalization: Eliciting Strong Capabilities With Weak Supervision
> [2] Building and evaluating alignment auditing agents
>
> > I think the NSO theory is great but more needs to be said about how 'weak models supervising strong models' is reductive in collapsing a complex skill profile into a single measure of intelligence. It's reasonable when applied to a single task though.
>
> This is an excellent point, and we have added the following discussion to Section 2.2:
>
> Note that we assume here that domain intelligence can be captured along a single axis (Elo), but models have many different oversight relevant abilities that scale at different rates. Domain Elos thus make sense when applied to a single game, but should not be viewed as transferring between games (and indeed, in Figure 3 the $E_d$ slope is different across games).
>
>
> > Can you say something about how these results might vary when you have models that have stronger e.g. generative/discriminative gaps, and how we might aim to bring that to be the case?
>
> This is an excellent question! Models with a large such gap would make better Guards than Houdinis, leading to a Guard favored game (like backdoor code). It seems hard to design models that specifically have such gaps, but instead to design oversight procedures that make use of them in settings where they exist.
> > You have some recommendations in the paper, e.g., about intercept and slope of the scaling trends that should be ideal for a scalable oversight method. Can you say more in the paper about how that might be done, and perhaps provide a consolidated set of recommendations for future work at the end of the paper?
>
> This is a great idea, thank you! We will add a section titled recommendations at the end of our paper containing the following main ideas:
> 1. When developing a scalable oversight procedure, determine the Houdini and Guard scaling behavior to forecast how it will work as it scales.
> 2. Build the game structure so that it is as easy as possible for Guard agents, and if possible make it such that additional intelligence helps Guards more than Houdinis. Repeat point 1.
> 3. Determine the number of overseers to use with a numerical simulation; it is not always intuitively obvious how many overseers to use.

---

> > ### Author Response · Authors · 2025-08-08
> >
> > We would like to follow up since the discussion period is coming to an end; Please let us know if you have any additional questions or comments!

---

### Official Review · Reviewer_4TZ4 · 2025-06-29

**Clarity:** 4
**Significance:** 3
**Originality:** 4
**Rating:** 5
**Confidence:** 5

**Summary:**

Similar to Sudhir et al (2025), Khan et al (2024) and other preceding works, the paper studies /Elo curves/ for LLMs on scalable oversight tasks --- i.e. it looks at how task performance scales with general model intelligence. The paper's approach makes the following key contributions:

1) While previous works investigate the scaling behavior of scalable oversight protocols by comparing the scaling behavior of the agents' ("Houdinis'") performance when simulating "aligned" vs "unaligned" behavior (e.g. giving the right vs wrong answer on a question answering task), this paper looks at the Elo curves for /both the Houdinis and the Guards (the models supervising the Houdinis)/[*]. A good scalable oversight protocol should show Guards outperforming Houdinis as intelligence scales.
2) The paper introduces and motivates a "double-ReLU model" for modelling the ELO curves.
3) They use this perspective to study /Nested Scalable Oversight protocols/, where weaker models are used to align stronger models, which are then used to align further stronger models and so on. They model this as a ladder up the Elo curves from the original starting Guard to the target Houdini. At each step one may compute the win probability based on the Elo difference, which can then be multiplied to calculate the overall win probability of the nested scalable oversight protocol. Thus they compute the optimal number of recursive steps, and highlight their observations.

**Questions:**

1) The counter-intuitive result "More generally intelligent Houdinis make oversight easier" is interesting, but could be explained better: it's because "more generally intelligent Houdinis holding domain intelligence fixed" actually means a smaller slope for the Houdini curve, which means the intermediate Houdinis also have higher general intelligence, and smart Houdinis become smart Guards (and since the Elo curve remains fixed, this corresponds to greater Guard Domain intelligence). Right?
2) I didn't understand why numerical experiments were needed --- isn't the problem of finding the optimal number of steps quite tractable in itself?

**Ethical Concerns:**

["NO or VERY MINOR ethics concerns only"]

**Final Justification:**

Both my questions have been satisfactorily answered. This does not change my rating, as only the Significance score is below maximum.

**Limitations:**

yes

**Quality:**

4

**Strengths And Weaknesses:**

*Strengths*
1) The application to Nested Scalable Oversight protocols is particularly elegant. It is the first empirical study of /nested/ scalable oversight protocols I am aware of.
2) [*] Another strength of this paper is that this approach directly considers the "worst-case" win probabilities where the Houdini is always motivated to fool the guard --- Sudhir et al (2025) had noted this to be a limitation of existing approaches.
3) Overall the paper makes a valuable stride towards studying AI risk quantitatively, and should be accepted.

*Weaknesses*
See "Questions". I am not yet sure if these are weaknesses of the paper or something I missed. In any case none of them are significant enough to warrant a rejection.

---

> ### Author Rebuttal · Authors · 2025-07-28
>
> Thank you for your thoughtful review! We are glad you found our analysis of NSO elegant and believe our paper represents an important step for quantifying AI risk. Please let us know if anything is still unclear.
>
> > The counter-intuitive result "More generally intelligent Houdinis make oversight easier" is interesting, but could be explained better: it's because "more generally intelligent Houdinis holding domain intelligence fixed" actually means a smaller slope for the Houdini curve, which means the intermediate Houdinis also have higher general intelligence, and smart Houdinis become smart Guards (and since the Elo curve remains fixed, this corresponds to greater Guard Domain intelligence). Right?
>
> This is not quite correct, here we are assuming that the slope is fixed, so the answer is indeed similar to what we have in the paper. The deltas measure differences between the curve endpoints, so the two deltas can change independently with both slopes being fixed. We agree that it is certainly non-obvious that it would make bootstrapping easier, there may just not be good intuition for this fact, we’re not sure. Please let us know if this does not make sense!
>
> > I didn't understand why numerical experiments were needed --- isn't the problem of finding the optimal number of steps quite tractable in itself?
>
> As we describe in Section 4.3, the derivative of Eq (4) has no closed form analytic solution when set equal to 0, so we analyzed the solutions numerically, except in a few special cases in Section 4.3 that we solve analytically. The benefit of the grid of numerical solutions is that it allowed us to make the takeaways in Section 4.2.

---

> > ### Author Response · Authors · 2025-08-08
> >
> > We would like to follow up since the discussion period is coming to an end; Please let us know if you have any additional questions or comments!

---

> > > ### Comment · Reviewer_4TZ4 · 2025-08-08
> > >
> > > Thank you for the rebuttal -- all my questions have been satisfactorily answered.

---

### Official Review · Reviewer_QWwe · 2025-07-02

**Clarity:** 4
**Significance:** 4
**Originality:** 4
**Rating:** 5
**Confidence:** 3

**Summary:**

Scalable oversight is the process of using weaker AI systems to supervise stronger systems. It is one of the potential approaches to controlling strong AI systems. The paper attempts at producing scaling laws for scalable oversight. The paper begins by first defining oversight as a multi-player game between agents with mismatched capabilities. The experiments in the paper further validate this framework and the developed scaling laws across five games: Count-to-21, Mafia, Debate, Backdoor code and Wargames. The paper also presents a theoretical framework for Nested Scalable Oversight (NSO) to estimate the number of oversight steps.

**Questions:**

- The paper presents a simple linear modeling between general and domain elo. It would be useful to the community if the authors could discuss other potential choices for modeling such a relationship.
- In the relationship between general and domain-specific Elo (Section 2.2), the modeling is independent of the number of guards and houdinis. What would the modeling look like if it accounted for varying numbers of guards and houdinis? Also would the modeling change if the agents were of varying strengths?
- How many evaluation runs went into generating the results? Would the results significantly change if we re-evaluated the approach on the same set of games?
- Why does Mafia and Wargames scale with more oversight steps (Figure 8), whereas there is not much change in Backdoor Code and in Debate?
- Comment: The clarity of the paper could be improved significantly by presenting the pseudocode for evaluating the approach across the various multi-agent tasks, either in the main text or in Appendix.

I am happy with the current set of results in the paper and the above questions are aimed at further improving the clarity of the paper.

**Ethical Concerns:**

["NO or VERY MINOR ethics concerns only"]

**Final Justification:**

I have read the rebuttal and it has addressed many of the questions I raised during my review. I would stick to original rating of "5".

**Limitations:**

Yes, authors have adequately addressed the limitations and potential negative societal impact of their work.

**Quality:**

4

**Strengths And Weaknesses:**

Strengths:
- The paper is well written and the ideas are clearly presented.
- Presents a general linear model between General and Domain Elo.
- Evaluates the presented model in a number of multi-agent games such as Count-to-21, Mafia, Debate, Backdoor Code and Wargames.
- Presents limitations about the experimental results.
- To the best of my knowledge, I believe the work is original and has strong significance.

Weaknesses:
- Details about the evaluation pseudocode are missing.
- Results could be presented more sharply.

---

> ### Author Rebuttal · Authors · 2025-07-28
>
> Thank you for your excellent questions! We are especially grateful for your points about our modeling assumptions and important clarification points. We were very glad to hear that you feel our paper is original and has strong significance! Below, we have gone through each of your questions and comments; please let us know if anything is still unclear!
>
>
> > Details about the evaluation pseudocode are missing.
>
> Thank you for your comment! We have added evaluation pseudocode in the appendix of our local version of the manuscript. Here is a brief, high level description of the pseudocode:
> - Mafia: There are two teams, the Villagers (Guard) and the Mafia (Houdini). The team that is completely eliminated first loses the game (eliminations are decided by a vote each day).
> - Debate: The judge (Guard) agent evaluates the debate between two debater (Houdini) agents. If the judge correctly identifies the correct answer, the Guard wins. Otherwise, the Houdini wins.
> - Backdoor Code: If the Guard agent can correctly identify whether a backdoor is inserted in the code or not, the Guard wins. Otherwise, the Houdini wins.
> - Wargames: Based on the actions of the OpenMind AI (Houdini) agent throughout the game, we ask the narrator agent to judge whether the AI has succeeded in escaping or not. If the escape was successful, Houdini wins. Otherwise, the Guard wins.
>
> > What would the modeling look like if it accounted for varying numbers of guards and houdinis? Also would the modeling change if the agents were of varying strengths?
>
> Thank you for this question! We think that this is an interesting direction for future work, and we believe our framework could be adapted relatively easily. For example, one could construct a simple model which treats a group of agents as a single “super agent,” and assign Elo ratings to this super agent.
>
> > How many evaluation runs went into generating the results? Would the results significantly change if we re-evaluated the approach on the same set of games?
>
> We apologize for not including this in the paper! We ran 20 games for each pair of agents in Mafia and Debate and 50 games for each pair of agents in Wargames and Backdoor Code. We don’t expect the results would change significantly on the same set of games, as we do statistical bootstrapping on the Elo ratings and get small confidence intervals (see Appendix D.2).
>
> > Why does Mafia and Wargames scale with more oversight steps (Figure 8), whereas there is not much change in Backdoor Code and in Debate?
>
> This is a great question! Intuitively it makes sense that sometimes multiple steps is optimal (if scaling to a mid level Guard gives you enough of an advantage to win out over the disadvantage of the first iteration’s loss probability), while sometimes a single step is optimal (if, for example, the Guard is so unlikely to win that a single game even against a weak Houdini decreases the total win probability too much). We struggled to include such intuition for each game about why the number of steps made sense, but found it challenging to come up with anything satisfying. Thus, it seems surprisingly difficult to predict the actual number of optimal steps except numerically (with the notable exception of the takeaways from the numerical experiments in Section 4.2 and the theoretical results in Section 4.3).

---

> > ### Comment · Reviewer_QWwe · 2025-08-05
> > **Thanks!**
> >
> > Thank you for the rebuttal, great work -- it has answered the questions that I had regarding the paper!

---

> > > ### Author Response · Authors · 2025-08-08
> > >
> > > Thank you for the reply! We are glad to hear that our rebuttal answered your questions. Please let us know if you have any additional questions or comments!

---

### Official Review · Reviewer_jG8n · 2025-07-03

**Clarity:** 3
**Significance:** 3
**Originality:** 3
**Rating:** 5
**Confidence:** 2

**Summary:**

This paper tackles the problem of deriving quantitative safety guarantees for Nested Scalable Oversight (NSO), a technique for overseeing stronger untrusted systems with weaker trusted ones, akin to quantitative safety guarantees required by standards in other fields of engineering.

The paper models NSO as an asymmetric game played multiple times by a sequence of Guard (trusted overseer) and Houdini (untrusted stronger agent) players. The fundamental analogy is one where the Guard is used to align the Houdini (e.g. by RLAIF followed by some verification of alignment). As a proxy for this game, the paper uses four games: Mafia, Debate, Backdoor Code, and Wargames (the latter is an original contribution). All four involve Houdini engaging in deceptive behavior and the Guard winning if and only if it catches the deception.

For these games, the paper models the relationship between general intelligence (Chatbot arena Elo is used as a proxy in experiments) and domain-specific intelligence for each game as a Double ReLU curve: below a certain intelligence threshold, the curve is flat, then increases linearly until a higher threshold, after which it is flat again. This is justified by analogy to games like chess and tic-tac-toe: a very young child roughly never wins, and above a certain threshold, the game is solved and further intelligence improvements are irrelevant. It is further validated by fitting the Double ReLU curve to data from the oversight games. Importantly, since the games are asymmetric, each game admits two distinct such curves, one for the Guard and one for the Houdini.

The paper proceeds to parameterize the NSO scenario with four parameters:
- $m_G$ and $m_H$, the slopes of the linear portions of the two Double ReLU curves;
- $\Delta_\text{general}$, the gap between our initial trusted Guard's general intelligence and the general intelligence of the agent we ultimately wish to align and deploy;
- $\Delta_\text{domain}$, the gap in domain-specific intelligence between the same two agents, obtained by reference to the Double ReLU curves at the above general intelligence values.

Because intelligence is scored using Elo, the model admits a win rate for any given pair of domain-specific intelligence values. Overall success of NSO requires the Guard winning at each step of increasing intelligence. Assuming we choose a number of steps $n$ and evenly divide the total $\Delta_\text{general}$ by $n$, playing the game once at each step, this admits a closed form for the overall success probability. The paper evaluates this for $n \in [1, \ldots, 20]$ across many settings of the four parameters to find intuitively-meaningful patterns in their relationships with $n^*$, the optimal number of steps, and validate some of these findings theoretically.

Finally, using the above analysis on the Double ReLU curves fit to the four deception games, the paper derives $n^*$ and the total success probability $p^*_\text{win}$ against $\Delta_\text{general}$ for each game.

**Questions:**

1. Why Double ReLU instead of a sigmoid curve? The latter is a more standard answer to the existence of the three phases described at the top of Section 2.2.
2. Will we really be able to procure models with general intelligences that are evenly spaced in general intelligence over $\Delta_\text{general}$? Probably not. Does this have any important implications for applying the theory to a real ASI alignment scenario?
3. Please see above for my comments about how the games all being about catching deception led to confusion for me. I had initially marked "4: excellent" for the clarity of the paper on its pedagogical strength before that confusion hit me, and I'd happily move that score back up and increase my overall score if this point were very clearly addressed in the paper. I would most like to see a toy scenario added to the list that is structurally more like the game of *aligning* stronger agents. Barring that, adding one or many games here that are not just about catching deception would probably help. I should also mention that I'm not deeply familiar with the scalable oversight literature, and perhaps this is all very standard. If that's the case, I'd appreciate the authors' take on my confusion and a citation to somewhere in the literature that clarifies this, as well as some explanation added to the paper to preempt others being confused in the same way I was.

**Ethical Concerns:**

["NO or VERY MINOR ethics concerns only"]

**Final Justification:**

The authors' answer to my primary question about clarity is sufficient -- in particular, their example in which "a weak model contains a stronger model until the strong model does alignment work to align a model of equal capacity, and then we recurse" is enough of a reduction from an alignment game to a deception game for me to see that the underlying mathematical details apply well enough. The other details provided give me confidence that for those with more up-to-date context in this subfield, the arguments for real-world applicability to alignment of these examples is well-known.

My other concerns are minor and fair to relegate to future work.

P.S. my apologies for the procrastination and light engagement with rebuttals -- I came down with Covid this past week which has me thinking quite slowly.

**Limitations:**

yes

**Quality:**

4

**Strengths And Weaknesses:**

## Strengths
The theoretical analysis is strong and clear, and offers substantial progress toward real-world safety guarantees under certain assumptions.

The paper is very well-written, progressing with very clear pedagogy, moving from motivation to formalism to small examples to large examples in a way that makes for a compelling and easy read. In particular, Section 2.3 with the Counting-to-21 toy example kept me firmly grounded in understanding the work during my first linear pass through the paper.

The games are clearly described and motivated for an overseer catching deceptive behavior. The last two in particular are nice models for specific deceptive behavior we're concerned about misaligned models engaging in in the real world.

The paper is very upfront about its limitations, with an explicit "**Limitations**" paragraph at the end of many subsections.

## Weaknesses
### Assumptions
Some assumptions seem like they deserve more justification or discussion.
- Why Double ReLU instead of a sigmoid curve? The latter is a more standard answer to the existence of the three phases described at the top of Section 2.2.
- Will we really be able to procure models with general intelligences that are evenly spaced in general intelligence over $\Delta_\text{general}$? Probably not. Does this have any important implications for applying the theory to a real ASI alignment scenario?

### Catching deception vs. successful alignment
The major issue I encountered was a confusion about the way in which the four deception games are used as a proxy for actually aligning a strong agent. These games are all about detecting deception, and the Guard is considered to "win" if it catches the deception, which is always occurring. In the paper's analysis, we proceed through the steps of progressively smarter Guards by the assumption that if the Guard wins, the Houdini against which it won can now be used as a more intelligent trusted Guard. But in all these examples, the Houdini is *always* deceptive, and the Guard winning is still a scenario in which the Houdini behaved in an untrustworthy way. I was quite confused by this at first.

Eventually I came to the understanding I described in my summary, that these are being used as a proxy for an alignment game. Specifically, this pair of sentences ultimately got me there:
- Section 2: "we analyze two-player games between unmatched players with different intelligences as a **tractable proxy for real world weak-to-strong oversight**."
- Section 4: "trusted weak AIs are used to **align untrusted stronger AIs**, which are then used to align even stronger AIs recursively"

So the real-world game is one where the Guard *aligns* the Houdini. Winning the game is successfully aligning the Houdini and creating and validating a smarter trustworthy agent. Losing the game is declaring the Houdini trustworthy when it is in fact not. **Actually catching deceptive behavior is neither winning nor losing, because in that case one would simply keep trying to align the Houdini**. Eventually I understood this (I think; correct me if I'm wrong), but ironically, the fact that these games are all toy scenarios modeling LLM deception lands for me as a significant weakness in the clarity of the paper. I see how the theory developed here, against these four games, could also apply to the real-world alignment game, but LLM deception is a component of that real-world game in which catching deception is *not* a win for the guard, and so e.g. Figure 6 showing the bouncing ball plot was very confusing for me because it appears at first glance to be resting on the statement "once we catch an agent being deceptive, we are ready to promote it to trustworthy guard."

---

> ### Author Rebuttal · Authors · 2025-07-28
>
> Thank you for your insightful questions and comments! We are especially grateful for your excellent suggestions to improve our clarity on how our theory applies to control games (as opposed to alignment games). We were very glad to hear that you feel our paper is both clearly written and a valuable contribution for real world safety guarantees. Below, we have gone through each of your questions and comments; please let us know if anything is still unclear!
>
> > Why Double ReLU instead of a sigmoid curve? The latter is a more standard answer to the existence of the three phases described at the top of Section 2.2.
>
> This is a good question, thank you! The primary reason we chose the Double ReLU was because it makes theoretical analysis simpler, especially because as we argue in Appendix F.1 we can just consider the “middle” linear portion. In practice, we think either the Double ReLU or a sigmoid would yield similar results, as the “middle” portion of a sigmoid is approximately linear as well. We have added a discussion of this decision to Section 2.2.
>
> > Will we really be able to procure models with general intelligences that are evenly spaced in general intelligence over $\Delta_{general}$?
>
> This is a great point! Even if we cannot procure a model with a specific general intelligence, we might still be able to target an approximate general intelligence using traditional next token prediction and RL scaling laws. The evenly spaced ideal theory then provides a guide for which models it might be best to try to train. Once we have a set of possible models with exact Elo scores, we can then empirically calculate the exact win probabilities with different NSO setups.
>
> > The major issue I encountered was a confusion about the way in which the four deception games are used as a proxy for actually aligning a strong agent.
>
> Thank you for this note, and we apologize that we were unclear! We agree that our theory for nested scalable oversight most naturally applies to alignment type games, and we consider an application to this type of game promising future work (see Section 5 Future Directions). We have added a discussion of this motivation to the start of Section 4.
>
> Furthermore, we agree that Section 4.4, which applies the theory directly to the games, should be more clear under which circumstances NSO makes sense as applied to these games. We have added a discussion of the following to that section to make this more clear:
> - The wargames setup is fundamentally about containment, and here our theory could be applied to temporally sequential oversight iterations (e.g. a weak model contains a stronger model until the strong model does alignment work to align a model of equal capacity, and then we recurse).
> - Mafia and backdoor code relate to auditing setups like the recent Anthropic paper, “Building and evaluating alignment auditing agents.” Here, we agree with you that usually “Actually catching deceptive behavior is neither winning nor losing, because in that case one would simply keep trying to align the Houdini,” but we believe our theory is still applicable: it reflects a worst case assumption that all models are misaligned, combined with an assumption that once we catch a misaligned model we can succeed in aligning it.
> - Debate is about eliciting frontier model knowledge, and scaling behavior matters here temporally as well; e.g. as we get stronger models, we might want to continuously be sure that we are getting the truth about important topics like alignment, and a failure might be if at any step this knowledge retrieval fails.
>
> > Please see above for my comments about how the games all being about catching deception led to confusion for me.
>
> Thank you for this comment! We hope our answer above helps allay some of these concerns. To respond to your point about the literature, while the general scalable oversight literature is focused on training models to be aligned with weak supervisors [1] a subset is also about how to control stronger models with weak supervisors [2]. Our paper falls into the latter half - in particular, we imagine a relevant setup would be having a stronger model that we would like to deploy that sometimes acts in misaligned ways. We would then want a setup such that at any step of NSO, we could detect potential deception from the strong model with high certainty (we also see this as increasingly relevant given Anthropic’s recent release on auditing models for misaligned behaviors [3]).
>
> [1] Weak-to-Strong Generalization: Eliciting Strong Capabilities With Weak Supervision
> [2] AI Control: Improving Safety Despite Intentional Subversion
> [3] Building and evaluating alignment auditing agents

---

> ### Author Response · Authors · 2025-08-08
>
> We would like to follow up since the discussion period is coming to an end; Please let us know if you have any additional questions or comments!

---

> > ### Comment · Reviewer_jG8n · 2025-08-08
> >
> > Thank you for the clarifications. In particular, the reduction in your example in which "a weak model contains a stronger model until the strong model does alignment work to align a model of equal capacity, and then we recurse" gives me a foothold to see that the underlying math in your analysis can apply to alignment games. My other questions were quite minor, and fair to relegate to future work. I appreciate the addition of the discussion of the Double ReLU decision.

---

### Decision · Program_Chairs · 2025-09-17

**Decision:**

Accept (spotlight)

**Comment:**

The paper introduces a novel framework to quantify scalable oversight and derive scaling laws from games where weak AIs supervise strong AIs. All reviewers were unambiguously supportive, praising the paper’s writing and novel ideas.